# Personalized Heterogeneous FL with Gradient Similarity

## Abstract

In the conventional federated learning (FL), the local models of multiple clients are trained independently by their privacy data, and the center server generates the shared global model by aggregating local models. However, the global model often fails to adapt to each client due to statistical and systems heterogeneities, such as non-IID data and inconsistencies in clients' hardware and bandwidth. To address these problems, we propose the Subclass Personalized FL (SPFL) algorithm for non-IID data in synchronous FL and the Personalized Leap Gradient Approximation (PLGA) algorithm for the asynchronous FL. In SPFL, the server uses the Softmax Normalized Gradient Similarity (SNGS) to weight the relationship between clients, and sends the personalized global model to each client. In PLGA, the server also applies the SNGS to weight the relationship between client and itself, and uses the first-order Taylor expansion of gradient to approximate the model of the delayed clients. To the best of our knowledge, this is one of the few studies investigating explicitly on personalization in asynchronous FL. The stage strategy of ResNet is further applied to improve the performance of FL. The experimental results show that (1) in synchronous FL, the SPFL algorithm used on non-IID data outperforms the vanilla FedAvg, Per-FedAvg, FedUpdate, and pFedMe algorithms, improving the accuracy by $1.81 \sim 18.46\%$ on four datasets (CIFAR10, CIFAR100, MNIST, EMNIST), while still maintaining the state of the art performance on IID data; (2) in asynchronous FL, compared with the vanilla FedAvg, Per-FedAvg, and FedAsync algorithms, the PLGA algorithm improves the accuracy by $0.23 \sim 12.63\%$ on the same four datasets of non-IID data.

## 1 Introduction

With the popularity of smartphones, personal computers, and other devices, the data stored on them has increased dramatically. These data are related to each other but exist independently on different devices. Also, their owners are often not willing to share their private data, which prompted Federated Learning (FL) (McMahan et al., 2017). Conventional FL is a distributed machine learning framework in which multiple clients train a global model together under a central server's coordination without sharing their local data. It's common process is as follows: (1) At each global iteration, the central server broadcasts its current global model to the client; (2) With the help of the global model, the client updates its local model which trained on its local data, and sends it back to the central server; (3) The central server aggregates the local models to get a new global model. The FL process will repeat these three steps until convergence. Under the premise of privacy protection, it can obtain a shared global model with higher generality. Nonetheless, the wide application of FL still is facing the following two major challenges:

- Statistical heterogeneity. As the user preference of each device may be different, the data distribution on each device may also be inconsistent. Consequently, the data as a whole on all the clients may be unbalanced and non-IID. This may cause undesirable performance of the FL participants.

- Systems heterogeneity. Due to the differences of users' hardware and network bandwidth, conventional FL models are prone to generate stragglers whose models are easy to be discarded after server's time-out.

Firstly, to address the statistical heterogeneity, the personalization of FL is widely studied. The majority of the available personalized FL (PFL) methods focus on one shared global model's parameters updating by client's fine-tune. FedAvg (McMahan et al., 2017) is treated by (Jiang et al., 2019a) as a client's meta-learning-like process fine-tuning on one shared global model with higher accuracy. FedMeta (Chen et al., 2018), also a personalized FL based on meta-learning, generates a local and personalized model for each client using a common global model. Different from the aforementioned methods, we propose a Subclass Personalized Federated Learning (SPFL) algorithm, which utilizes the Softmax Normalized Gradient Similarity (SNGS) to generate global models with each specific to one client for its parameter fine-tune.

Secondly, to tackle the systems heterogeneity, the server of an asynchronous FL algorithm named FedAsync (Xie et al., 2019), can immediately perform aggregation after receiving a local model, reducing the waiting time for stragglers. In TWAFL (Chen et al., 2019), a time-weighted aggregation strategy is used by server to handle stragglers according to the staleness of the received model parameters. Although these asynchronous FL algorithms can overcome some difficulties encountered in synchronous aggregation, such as waiting for the response from client, they are still inefficient for directly aggregating the received local model on the server, without taking the iteration gap of stragglers' local models at global communication round into consideration. Therefore, we propose the Leap Gradient Approximation (LGA) algorithm using SNGS to predict the local model of the stragglers in each global communication round. These local models estimated by server are also aggregated with non-stragglers' local models on the server to attained one shared global model. Further, to consider the real scenarios where system heterogeneity and statistical heterogeneity often co-exist, personalization is employed in our asynchronous FL. Before each individual personalized model for client is dispatched from server, it is computed by using the shared global model and the SNGS value between server and that client. We refer this personalized asynchronous FL approach as Personalized Leap Gradient Approximation (PLGA) algorithm.

**Contribution**   The main contributions of this paper towards personalized heterogeneous FL can be summarized as follows: (1) We propose the SPFL algorithm for both IID and non-IID data in personalized synchronous FL, where the server uses the SNGS to weight the relationship between clients, and delivers the personalized global model to each client; (2) We propose the PLGA algorithm for the personalized asynchronous FL, in which the server also applies the SNGS to weight the relationship between client and itself, and uses the first-order Taylor expansion of gradient to approximate the model of the delayed clients; (3) The stage strategy of ResNet is applied to improve the performance of both SPFL and PLGA algorithms.

## 2   RELATED WORK

In general, the personalized process in FL can be achieved by a variety of ways. The work (Hard et al., 2018) shows that by adding user-related features into the local inference process to get a personalized model may be better than the global one. The completely decentralized algorithms for each client by smoothing some similar tasks or model parameters between similar data distributions are studied in (Bellet et al., 2018; Vanhaesebrouck et al., 2017). Algorithms are also proposed to achieve personalization by jointly learning the model on a similar graph structure (Lalitha et al., 2019; He et al., 2021). FedFOMO (Zhang et al., 2020) is another approach to PFL, where each client only federates with other relevant clients for model aggregation. The pFedHN (Shamsian et al., 2021) algorithm is proposed using hypernetwork to generate different personalized model for each client.

Furthermore, PFL can also be studied by its relationship with other machine learning paradigms. The close relationship between FL and meta-learning is analyzed by (Khodak et al., 2019; Jiang et al., 2019b). Following the classic meta-learning algorithm named MAML (Finn et al., 2017), Reptile algorithm (Nichol et al., 2018) is proposed , which has a close relationship with the Federated Averaging algorithm. When each individualized model learning is treated as a task, such a scenario can be viewed as a multi-task learning(Zhang & Yang, 2017), in which each task learns a task-related model. MOCHA (Smith et al., 2017) is designed by regarding FL as a multi-task, which solves communication efficiency and fault tolerance. To provide a good global model, there are PFL applying transfer learning and domain transfer (Mansour et al., 2009; Cortes & Mohri, 2014; Ben-David et al., 2010). Model mixing is another strategy to approach PFL (Deng et al., 2020; Hanzely

& Richtárik, 2020; Liang et al., 2020). Alternatively, Other PFL methods apply clustering (Mansour et al., 2020; Ghosh et al., 2020; Duan et al., 2020). Finally, pFedMe (Dinh et al., 2020) applies Moreau Envelopes as clients' regularized loss functions to achieve personalization.

There are just few studies on asynchronous FL. TWAFL (Chen et al., 2019) aggregates a temporally weighted local models on the server. FedAsync (Xie et al., 2019) introduced an algorithm to balance the parameters of the updated model and the last global model. Meanwhile, the distillation technique has also been applied in (Li & Wang, 2019; Bistritz et al., 2020).

## 3 METHOD

Our method generates a personalized model for each client during the server aggregation process. Therefore, the initialization model of each client is personalized. At the same time, we also consider the contribution of different neural network layers to the personalized model. We will introduce the application of this personalized algorithm to synchronous FL and asynchronous FL in the following subsections.

---

**Algorithm 1** Subclass Personalized Federated Learning (SPFL) Algorithm

---

**Input:** initialized global model parameters $w^0$; initialized SNGS matrix $\tilde{S}_s \in R^{N \times N}$; initialized personalized local model of each client's $\{w_i^0\} = w^0$; update frequency $\Gamma$ and the number S of SNGS matrix; client learning rate $\beta$; server learning rate $\alpha$; local epoch E; local batch size B of each client

**Output:** Model Parameters $\{w_i\}$

**Def** *MainLoop*:
  **for** $t = 0, 1, \ldots$ **do**
    perform the following steps in parallel for each client $C_i \in \mathcal{C}$
    **if** $t\%\Gamma == 0$ **then**
      $w_{gi}^t = \frac{1}{N} \sum_i^N w_i^t$

      $\{g_i^t\}$ = ClientUpdate $(w_{gi}^t, C_i, \beta)$

      $\tilde{S}_s$ = Update$\tilde{S}$Matrix $(\{g_i^t\})$
    $\{g_i^t\}$ = ClientUpdate $(w_{gi}^t, C_i, \beta)$
    $\{w_{si}^{t+1}\}$ = ServerUpdate $(\{w_{gi}^t\}, \{g_i^t\}, \tilde{S}_s, \alpha)$

**Def** *ClientUpdate($w_{gi}^t, C_i, \beta$)*:
  $w_i^t = w_{gi}^t$
  **for** $e = 0; e < E; e++$ **do**
    **for** *batch size b in B* **do**
      $w_i^t \leftarrow w_i^t - 2\beta \nabla L_i(f(w_i^t, x_b), y_b)$
  return $\{w_{gi}^t - w_i^t\}$

**Def** *Update$\tilde{S}$Matrix ($\{g_i\}$)*:
  **for** $s \in S$ **do**
    perform (3) and (6) to return $\tilde{S}_s(i, j)$

**Def** *ServerUpdate ($\{w_{si}^t\}, \{g_i^t\}, \tilde{S}_s, \alpha$)*:
  **for** $s \in S$ **do**
    perform (7) to get $w_{si}^{t+1}$

  return $\{w_{gi}^{t+1}\}$

---

**Definition of personalized FL algorithm** In FL, there is a set of $N$ clients $\mathcal{C} = \{C_i\}, i = 1, \ldots, N$. Define the dataset on all clients as $\mathcal{D} = \{D_i\}, i = 1, \ldots, N$, with $D_i$ as the corresponding dataset of $C_i$. The sample size of each datasets $D_i$ is $|D_i|$. $(x_j, y_j)$ is a sample of $D_i$, with $x_j \in R^d$ as the corresponding input feature and $y_j \in R^t$ as the corresponding input label. The prediction model of

client $C_i$ defined as $\hat{y}_j = f(w_i, x_j)$, where $w_i \in R^z$ is the model parameter. The dataset on each client $C_i$ is assumed to obey a distribution of $P_i$. And the loss function on each client $C_i$ is $L_i(w_i)$. The optimization goal for each client is:

$$\underset{w_i}{\arg\min}\, L_i(w_i) = \mathbb{E}_{D_i \sim P_i}[L_i(f(w_i, x_j), y_j)] \approx \frac{1}{|D_i|} \sum_{j}^{|D_i|} L_i(f(w_i, x_j), y_j). \qquad (1)$$

## 3.1 Subclass Personalized FL (SPFL) algorithm

We compared the performance of each client's independent training (i.e., the model in each client performs SGD only on local data available, and model averaging is not performed ) and conventional global FL algorithms, such as FedAvg. The results in Appendix B. show that FL usually outperforms each client's independent training on IID data, demonstrating the global model trained with multiple client data improves the model generalizability ability of each client. On the other hand, for some clients, independent training has better performance on non-IID data. Our experiment in Appendix B illustrates that the global FL model cannot adapt well to the heterogeneous data. To tackle this problem, we propose a personalized FL algorithm that improved the performance of conventional global FL algorithms on non-IID data. At the same time, it still maintains a high performance on IID data.

Firstly, we introduce a similarity matrix $S \in R^{N \times N}$ to model the relationship between different clients, with $S(i, j)$ represents the similarity between $C_i$ and $C_j$. If the data distributions $P_i$ and $P_j$ are similar, then the value of $S(i, j)$ is relatively high, and vice versa.

For any two clients, $C_i$ and $C_j$, supposing they receive the global model parameters $w^t$ at the same time $t$. After the local update of both clients using $w^t$, their local model parameters are $w_i^t$ and $w_j^t$, respectively. Then the SGD updates of the corresponding gradients of $w_i^t$ and $w_j^t$ are:

$$\begin{aligned} g_i^t &= w^t - w_i^t, \\ g_j^t &= w^t - w_j^t. \end{aligned} \qquad (2)$$

The correlation between $C_i$ and $C_j$ based on the global model $w^t$ is measured by the cosine similarity of the gradient updates $g_i^t$ and $g_j^t$, as calculated in (3):

$$S(i, j) = \frac{{g_i^t}^T g_j^t}{\|g_i^t\| \|g_j^t\|}, \qquad (3)$$

where $\| \cdot \|$ is the vector $L_2$ normal. And the aggregation strategy on the server for $C_i$ is as follows:

$$w_i^{t+1} = w_i^t - \alpha \sum_{j}^{N} \frac{|D_j|}{|D|} S(i, j) g_j^t, \qquad (4)$$

where $\alpha$ is the learning rate of the server. As the gradient itself is extremely directionality sensitive, when $S(i, j)$ is negative, multiplying it with the original gradient $g_j^t$ will reverse the gradient direction. Therefore, softmax function is applied to make $S(i, j)$ nonnegative and normalized as $\tilde{S}(i, j)$ . $\tilde{S}(i, j)$, the softmax version of similarity relationship between $C_i$ and $C_j$, is defined as:

$$\tilde{S}(i, j) = \frac{e^{S(i,j)}}{\sum_{j}^{N} e^{S(i,j)}}. \qquad (5)$$

The experimental results in Appendix B show that different layers influence personalization differently in the convolutional neural network (CNN). Inspired by ResNet, we use stage strategy that contains a set of network layers to measure the similarity between clients. Therefore, the similarity matrix $\tilde{S}$ is calculated as $\tilde{S}_s$ based on a stage as in (6), where its subscript $s$ indicates a stage:

$$\tilde{S}_s(i, j) = \frac{e^{S_s(i,j)}}{\sum_{j}^{N} e^{S_s(i,j)}}. \qquad (6)$$

Finally, the aggregation strategy on the server for $C_i$ base on stage is defined as:

$$w_{si}^{t+1} = w_{si}^t - \alpha \sum_j^N \frac{|D_j|}{|D|} \tilde{S}_s(i,j) g_{sj}^t, \tag{7}$$

where $\alpha$ is the learning rate of the server, $g_{sj}^t$ and $w_{si}^t$ are the updated gradient and model parameters of the stage $s$ on the client $i$ at iteration round $t$, respectively.

As the model changes with iteration round $t$, so does the gradient of the model. Consequently, the similarity matrix $\tilde{S}$ obtained based on the gradient will also change. Therefore, the similarity matrix is updated regularly during the training process. During each update, the model parameters of all clients are aggregated with equal weight to get $w_t$ of (2), and the similarity matrix is calculated based on $w_t$. In the view of each aggregation, it is still similar to the federated average algorithm. However, each client obtains a personalized model based on its similarity matrix in the training rounds in each aggregation. In this process, each client is individually regarded as a subclass, and the algorithm is called Subclass Personalized Federated Algorithm (SPFL). The algorithm is shown in Alg. 1.

### 3.2 PERSONALIZED LEAP GRADIENT APPROXIMATION (PLGA) ALGORITHM

This section introduces an asynchronous updating strategy that addresses the system heterogeneity problem mentioned in Section . The server can perform model parameters aggregation immediately after receiving few local models from clients within a small time window, overcoming the inefficiency of the conventional synchronous updating strategy, which either updates the model parameters after all the clients uploads their local model or discards the stragglers.

---

**Algorithm 2** Personalized Leap Gradient Approximation (PLGA) Algorithm

---

**Input:** Initialized global model parameters $w^0$; local step $E$; local batch size B of each client; client learning rate $\beta$; server iteration round $T$; client number $N$

**Output:** Model parameters $\{w_{pi}^t\}$ and $\{w^t\}$

**Def** *MainLoop* $(\{w_i^t\}, w_k^t)$**:**

    **for** $t = 0; t < T; t{+}{+}$ **do**

        ClientUpdate($w^t$,$C_k$,$\beta$)

        The server receives multiple local models $\{w_i^t\}_{i=0}^{N'}$ at the same time or within a small time window

        for the stragglers $w_k^t$, do (8), (13), (11), (14) in sequence;

        the server aggregates all clinets' model:$w^{t+1} = \frac{1}{N'} \sum_{i=0}^{N'} w_i^t$

**Def** *ClientUpdate* $(w^t,C_k,\beta)$**:**

    $w_k^{t-1} = w^t$

    **for** $e = 0; e < E; e{+}{+}$ **do**

        **for** $b \in B$ **do**

            $w_k^t \leftarrow w_k^{t-1} - \beta \nabla L_k(f(w_k^{t-1}, x_b), y_b)$

    return $w_k^t$

---

**Definition of asynchronous FL** Assuming that the FL process consists of $T$ global iterations that the number of the server aggregation. During each global iteration $s$, after receiving a global model parameters $w^{t_s}$, $0 \leq s \leq T$, from the server, the client $C_i$ uses the local data $D_i$ for a local training of $E$ epochs. After finishing the local training, each client uploads its local model $w_i^{t_s}$ to the server for aggregation. Then the next global iteration is performed. On the other hand, as shown in Fig. 1 for the conventional asynchronous FL process, the model parameters $w_k^{t_o}$ and $w_i^{t_\tau}$ uploaded by different clients $C_k$ and $C_i$, respectively, have equal importance in the server aggregation round $\tau + 1$, ignoring the fact that model $w_i^{t_\tau}$ performs several more SGD (local updates) than model $w_k^{t_o}$ that is called stragglers. Intuitively, as the server iteration round increase,the gap between server and stragglers get bigger, the adverse impact on server aggregation increases.

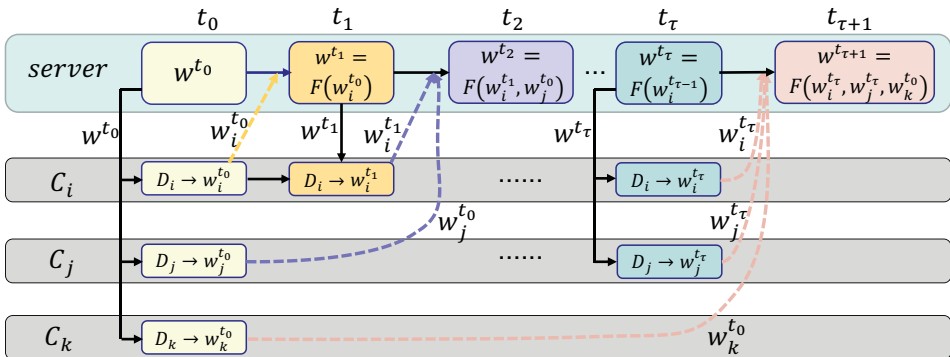

Figure 1: The illustration of asynchronous FL. The straggler $C_j$ need more computation time, and the straggler $C_k$ has a poor network bandwidth. The server aggregates models immediately and does not need to wait for all clients in asynchronous FL. $F$ is a specific aggregation strategy, $t_\tau$ is time of the $\tau$-th server aggregation round.

**Leap Gradient Approximation (LGA)** Here, we let the server aggregation round $s = 0$ to simplify introduction. First, the client $C_k$ receives the global model $w^{t_0}$ delivered by the server at the global iteration round 0 for initialization, and the $w_k^{t_0}$ is the local model of client $C_k$ that is trained by local data $D_k$. $w^{t_1}$ is the global model obtained right after the server performing an aggregation at iteration round 1. So the gradient similarity $S(k)$ of the server and the client $C_k$ can be calculated as follows:

$$S(k) = \frac{\left(w^{t_1} - w^{t_0}\right)^T \left(w_k^{t_0} - w^{t_0}\right)}{\left\| w^{t_1} - w^{t_0} \right\| \left\| w_k^{t_0} - w^{t_0} \right\|}. \tag{8}$$

We also apply the softmax function to normalize $S(k)$.

$$\tilde{S}(k) = \frac{e^{S(k)}}{e^1 + e^{S(k)}}. \tag{9}$$

As shown in as in Fig. 1, $w_k^{t_0}$ is directly used to compute the global model $w^{t_{\tau+1}}$ in conventional asynchronous FL. To avoid this, assuming that the training loss function is $L_1$-$Lipschitz$, inspired by DC-ASGD (Zheng et al., 2017), to alleviate the update gap between stragglers and the server, we use the approximation of the first-order Taylor expansion of the gradient to estimate the model of straggler $C_k$ in the current server iteration round $\tau$ by (10):

$$w_k^{t_\tau} = w^{t_0} + \tilde{S}(k) \left(w_k^{t_0} - w^{t_0}\right) \odot \left(w_k^{t_0} - w^{t_0}\right) \odot \left(w^{t_\tau} - w^{t_1}\right) + \left(w_k^{t_0} - w^{t_0}\right), \tag{10}$$

where $w_k^{t_\tau}$ is the local model of straggler client $C_k$ at the server aggregation round $\tau$, and $\odot$ is the element-wise product. The last two terms on the RHS of (10) is the prediction of the changes of $w_k^{t_0}$, which we called as gradient leap; especially, $Diag\left(\left(w_k^{t_0} - w^{t_0}\right) \odot \left(w_k^{t_0} - w^{t_0}\right)\right)$ is the approximation of the gradient of the gradient of $w_k^{t_0}$ . The first term on the RHS is the initial value of $w_k^{t_0}$. Furthermore, we also proposes another view on calculating $w_k^{t_\tau}$ as in (11)

$$w_k^{t_\tau} = w^{t_\tau} + \tilde{S}(k) \left(w_k^{t_0} - w^{t_0}\right) \odot \left(w_k^{t_0} - w^{t_0}\right) \odot \left(w^{t_\tau} - w^{t_1}\right) + \left(w_k^{t_0} - w^{t_0}\right), \tag{11}$$

where $w^{t_\tau}$ is the global model at the server aggregation round $\tau$. The only difference between (10) and (11) is their first terms on their RHS. In (11), it says that we want to predict $w_k^{t_\tau}$ based on current global model $w^{t_\tau}$, yet still use the prediction of the changes of $w_k^{t_0}$ in the previous FL iteration round.

**Personalized Leap Gradient Approximation (PLGA)** The personalized model for each client $C_k$ in the server aggregation round $\tau$ is as follows:

$$w_{kp}^{t_\tau} = w^{t_0} + \left(1 - \tilde{S}(k)\right) \left(w_k^{t_\tau} - w^{t_0}\right) + \tilde{S}(k) \left(w^{t_\tau} - w^{t_0}\right). \tag{12}$$

where $\tilde{S}$ and $w_k^{t_\tau}$ are derived from (9) and (10) or (11), respectively. Similar to (6), we can also apply stage strategy to $\tilde{S}(i)$, and obtain:

$$\tilde{S}_s(k) = \frac{e^{S_s(k)}}{e + e^{S_s(k)}},\tag{13}$$

and

$$w_{skp}^{t_\tau} = w_s^{t_0} + \left(1 - \tilde{S}_s(k)\right)\left(w_{sk}^{t_\tau} - w_s^{t_0}\right) + \tilde{S}_s(k)\left(w_s^{t_\tau} - w_s^{t_0}\right).\tag{14}$$

The personalized leap gradient approximation (PLGA) algorithm is introduced in Alg. 2.

## 4 EXPERIMENTS

In this section, we first evaluate the performances of SPFL compared with FedAvg(McMahan et al., 2017), Per-FedAvg(Fallah et al., 2020), FedUpdate(Jiang et al., 2019a) and pFedMe(Dinh et al., 2020) when the data distributions are IID and non-IID, we also explore the effect of hyperparameters $k, N, E, T$. We then compare PLGA with FedAvg (both synchronous and asynchronous settings), Per-FedAvg(Fallah et al., 2020), FedAsync(Xie et al., 2019) when the data distributions are non-IID. The source code is available at `https://github.com/MondayCat/PLGA`

**Experimental Settings**  For both of SPFL and PLGA experiments, we use four datasets, namely, CIFAR10 (Krizhevsky et al., 2009), CIFAR100 (Krizhevsky et al., 2009), MNIST (Lecun et al., 1998), EMNIST (Cohen et al., 2017) datasets. These datasets are suitable for testing different real-world scenarios in the proposed FL framework, such as non-IID, balanceless datasets. We run the experiments with 10 clients and a server, where each client has access to 6,60,6,32 classes of the above datasets, respectively. The detailed introduction and construction of the datasets can be found in Appendix B. All datasets are split randomly with 80% and 20% for training and testing, respectively. All experiments were conducted using PyTorch (Paszke et al., 2019) version 1.1.0. All the baselines have the same parameters and network structures as our model. We use 10 seeds to get a average result for each method.

### 4.1 SPFL EVALUATION

In the synchronous FL, we compare the proposed SPFL with the FedAvg, FedUpdate, Per-FedAvg , respectively. Although the FedAvg algorithm is not a personalized learning algorithm, its experimental results can still be helpful as a referances. FedUpdate and Per-FedAvg are FL algorithms dedicated to personalization based on the meta-learning. And pFedMe is using Moreau Envelopes to get a PFL model. The common parameters are fixed as $B = 128$, $\alpha = \beta = 0.01$, $\Gamma = 10$, $E = 1$, $S = 2$. We provide an extensive ablation study on design parameters' choices in Appendix C. We also test the personalized learning based on weight fusion, and the aggregation strategy changes from $w_i = w_i + \alpha\frac{1}{M}\sum_j \tilde{S}_s g_j$ to $w_i = \frac{1}{M}\sum_j \tilde{S}_s w_j$. The weight-based personalized SPFL algorithm is defined as SPFL-w. Noting that twice local updates, which mimic MAML, are used in SPFL, to explore the influence of the number of local updates on the experiment, we also add two models using once local update on the clients as ablation studies: SPFL-w (1-step) and SPFL (1-step).

As shown in Table 1, the accuracies on the four datasets are quite different, and decreasing in the order as MNIST, EMNIST, CIFAR10, CIFAR100. It also reflects the difficulty of each dataset. On a whole, the handwritten number recognition is relatively simpler than the image classification. And the two datasets within the same category are also different, such as MNIST and EMNIST, or CIFAR10 and CIFAR100. For all methods evaluated, the accuracy on IID dataset is always higher than that of non-IID. Therefore, the evaluation results on the four datasets are relatively objective and generalized, and not overfitting to one dataset.

SPFL performs much better on four non-IID datasets, compared to FedAvg, FedUpdate and Per-FedAvg. For IID datasets, Per-FedAvg has the best results except on EMNIST dataset. Nonetheless, the performance gap between SPFL and the best methods is still acceptable on IID datasets. The reason that SPFL performs a little bit worse on IID datasets is due to the delay of the gradient similarity calculation. Considering the actual application scenarios of personalization, in which the possibility of IID data on the client is relatively small, our algorithm can bring more benefit in the real environment.

Table 1: Comparison of clinet average accuracy (%) of SPFL and baseline algorithms on IID datasets and non-IID datasets. The best and second to the best performance are bolded and underlined, respectively.

| Methods | MNIST | | CIFAR10 | | EMNIST | | CIFAR100 | |
| --- | --- | --- | --- | --- | --- | --- | --- | --- |
| | IID | non-IID | IID | non-IID | IID | non-IID | IID | non-IID |
| FedAvg | 98.46±0.21 | 96.52±0.21 | 67.00±3.38 | 48.76±1.10 | 90.41±1.85 | 84.42±0.35 | 27.39±0.97 | 18.79±1.04 |
| FedUpdate | 98.46±0.22 | 97.21±0.35 | 66.45±3.37 | 59.45±1.22 | 89.95±1.88 | 85.80±0.67 | 26.61±1.08 | 21.13±0.86 |
| Per-FedAvg | 98.89±0.15 | 97.19±0.42 | 74.00±3.04 | 55.65±1.41 | **91.27±1.84** | 85.32±0.41 | **35.03±1.21** | 24.79±1.09 |
| pFedMe | 98.80±0.16 | 97.18±0.22 | 72.82±3.15 | 55.47±3.39 | 91.03±1.83 | 85.30±0.68 | 33.93±0.76 | 24.20±1.33 |
| SPFL-w (1-s) | 98.46±0.18 | 96.52±0.23 | 67.03±3.44 | 48.96±1.07 | 90.32±2.13 | 83.23±3.48 | 27.49±0.89 | 18.87±0.93 |
| SPFL-w | **98.90±0.16** | 97.22±0.36 | **74.04±3.18** | 55.36±1.14 | 90.91±2.11 | 85.25±0.38 | 34.90±1.10 | 24.73±0.99 |
| SPFL (1-s) | 98.35±0.16 | 97.71±0.16 | 65.53±4.24 | 61.36±0.84 | 89.94±2.85 | **87.66±1.84** | 25.79±0.99 | 22.08±0.87 |
| SPFL | 98.82±0.16 | **98.33±0.16** | 71.91±3.56 | **67.22±1.15** | 89.53±2.30 | 87.49±0.72 | 28.52±1.01 | **24.97±1.00** |

Compared to FedAvg and independent training of each client in Appendix B, SPFL outperforms both no matter on IID datasets or non-IID datasets. To differentiate the influence of gradient and weight aggregation strategy, we observed that SPFL-w is slightly better in the case of IID data, but SPFL is much better than SPFL-w on non-IID data. Nonetheless, SPFL-w outperforms the FedAvg and the independent training model both on IID and non-IID datasets.

The expermental results also show that the twice local updates dominate its counterpart, which indicate that MAML alike updates is helpful in personalization than once local update.

## 4.2 PLGA EVALUATION

In the asynchronous FL, the proposed PLPA is compared with the FedAvg (in the synchronous and asynchronous conditions), FedAsync and Per-FedAvg. Although the FedAvg and Per-FedAvg are synchronous federated algorithms, their experimental results can still be helpful as references here. FedAvg(Sync) means that discard the straggers when server aggregate, while FedAvg(Async) means the straggers participate directly in server aggregation. FedAsync is an FL algorithm dedicated to asynchronous FL. The hyperparameters are fixed as $B=64$, $\beta=0.001$, $E=20$, $T=800$. Based on ablation experiments, we find that these parameters have best performance. We use $N=10$ clients to evaluate the performance of the PLGA algorithm. To make the simulation of the asynchronous environment simple and controllable, we fix the set of straggler of each asynchronous FL experiment as $\mathcal{S}$, and the rest clients in set $\mathcal{C} \setminus \mathcal{S}$ participate in each server iteration round synchronously. For stragglers in $\mathcal{S}$, we let them missing server aggregation round periodically with different fixed missing period as $1, 2, \ldots, |\mathcal{S}|$, where the values correspond to the number of server iteration rounds. Suppose that the fixed missing period of the straggler in $\mathcal{S}$ is 2, the straggler will participate one server iteration and miss two server iterations, and then participate one server iteration and miss two server iterations again, and so on. Here, we use non-IID dataset (see Appendix B) for the evaluation as it is the most common data case in asynchronous FL. The experimental results in Table 2, where the total number of the stragglers is $|\mathcal{S}|=5$, show that the clients' performances of PLGA on the four non-IID datasets have a notable advantage over the other baseline algorithms. Note that when $|\mathcal{S}|=0$, it is equivalent to synchronous FL. We also notice that PLGA is better than LPA as a result of personalization, and that $w_k^{t_\tau}$ computed by (11) is better than by (10) for PFGA.

Table 2: Comparison of test accuracy (%) of the PLGA and baseline algorithms on non-IID datasets. The best and second to the best performance are bolded and underlined, respectively.

| Methods | MNIST | | CIFAR10 | | EMNIST | | CIFAR100 | |
| --- | --- | --- | --- | --- | --- | --- | --- | --- |
| | client | server | client | server | client | server | client | server |
| FedAvg (Async) | 97.24± 0.32 | 97.61±0.12 | 42.09±2.07 | 46.52±2.63 | 58.10±3.68 | 74.05±2.53 | 18.53±1.11 | **20.98±0.93** |
| FedAvg (Sync) | 97.31±0.39 | 97.72±0.18 | 42.36±1.96 | **46.86±2.24** | 61.05±4.00 | 74.22±2.97 | 19.38±1.00 | 20.98±1.17 |
| Per-FedAvg | 97.27±0.35 | 97.68±0.15 | 41.79±2.03 | 45.48±2.59 | 55.55±3.43 | 73.74±2.61 | 18.12±1.07 | 20.71±0.94 |
| FedAsync | 97.18±0.44 | 97.59±0.16 | 43.59±2.63 | 46.51±2.27 | 61.09±3.39 | **74.74±2.59** | 18.63±0.98 | 20.53±1.01 |
| LGA | 97.48±0.28 | 97.82±0.15 | 44.06±3.21 | 45.73±3.68 | 62.02±2.73 | 73.93±2.71 | 19.41±1.09 | 20.74±1.15 |
| PLGA | **97.83±0.16** | 97.82±0.16 | **46.11±3.22** | 45.96±3.59 | **68.18±2.31** | 73.91±2.72 | **20.73±1.15** | 20.84±1.12 |

# 5 CONCLUSION AND FUTURE WORK

This paper studied the personalized heterogeneous FL with gradient similarity. We tackled the statistical and system heterogeneity in FL in two scenarios, namely synchronous FL and asynchronous FL with SPFL and PLGA algorithms, respectively. In both SPFL and PLGA algorithms, SNGS is proposed to capture the similarity between the participants of FL, ether among clients or between sever and client. Stage concept is also studied in the computing of SNGS. The experiment results demonstrate that the proposed algorithms have excellent performances on non-IID data, while still maintaining the state of the art performance on IID data. We are going to further work on the theoretical proof of the convergence of the proposed algorithms, and also the sampling strategies in the evaluation of SNGS between clients to alleviate the communication and computation overheads.

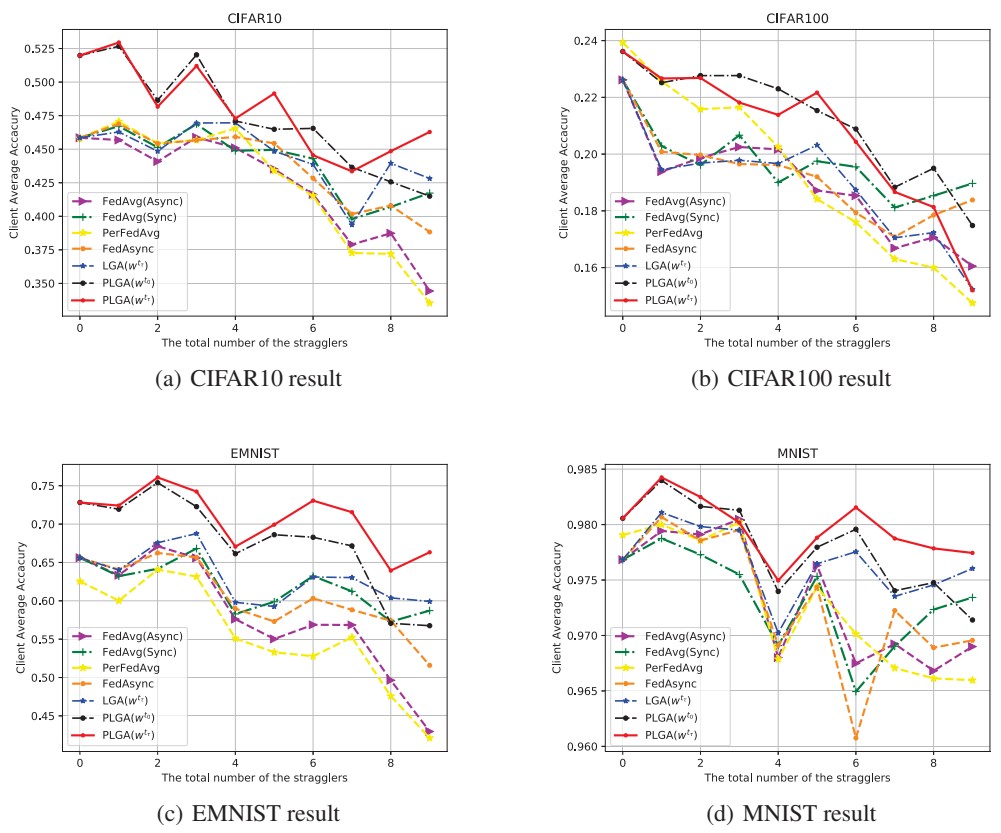

(a) CIFAR10 result

(b) CIFAR100 result

(c) EMNIST result

(d) MNIST result

Figure 2: The average accuracy results of the 10 clients of PLGA algorithm on the four datasets

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

## A  APPENDIX

## B  PREPARATION OF IID AND NON-IID DATASETS

Here we choose four public benchmark datasets MNIST, CIFAR10, CIFAR100, and EMNIST, as the original data source to construct the IID and Non-IID datasets. A brief introduction to the four datasets is shown in table 3. It should be noted that all non-IID situations in this paper are feature distribution skewed, that is, while the distribution $P(X)$ of feature $X$ on each client may be different, the distribution of conditional probability $P(Y|X)$ of label $Y$ is the same. Specifically, the sampling steps designed in this paper are as follows: Assuming that the original dataset is $D$, the total number of classes in the dataset is $K$, and the dataset of class $k$ is $D^k$. Supposing the total number of clients participating in personalized FL is $N$, for client $c$, the actual number of classses it has is $k_c, 2 \leq k_c \leq K$. The steps of preparing IID and non-IID datasets are as follows:

1. For any class $k$, its dataset $D^k$ is randomly and equally partitioned into $N$ parts, where $N$ is the total number of clients. Dataset $D_n^k$ denotes the $n$-th partition portion of $D^k$. For the generation of IID dataset goto step 2, otherwise goto step 3.

| | |
|---|---|
| MNIST: | MNIST is a handwriting recognition dataset, including 0-9, 10 kinds of handwritten characters. The training set contains 60,000 images, and the test set contains 10,000 images. Each image is standardized, the size of each image is 28*28, and the image is a grayscale image. |
| EMNIST: | The EMNIST dataset can be regarded as an extended version of the MNIST dataset. The dataset includes 671,585 numbers and letters in upper and lower case images in 62 categories. There are 47 categories except for the case. |
| CIFAR10: | The CIFAR10 dataset includes 60,000 32*32 color images in 10 categories, with 6000 images in each category. The training set is 50,000 sheets, and the test set is 10,000 sheets is uniformly distributed for each category of the training set and the test set. |
| CIFAR100: | The format of this dataset is the same as CIFAR10; the difference is that it contains a total of 100 classes of images, and each class has a capacity of 600 images. The total number of datasets is also 60,000, and the training data is 50,000 sets, and the test data is 10,000. Unlike the datasets of CIFAR10, because CIFAR100 has more types of data, it can have a higher degree of freedom in constructing a simulated FL data distribution. |

Table 3: Overview of the original four datasets

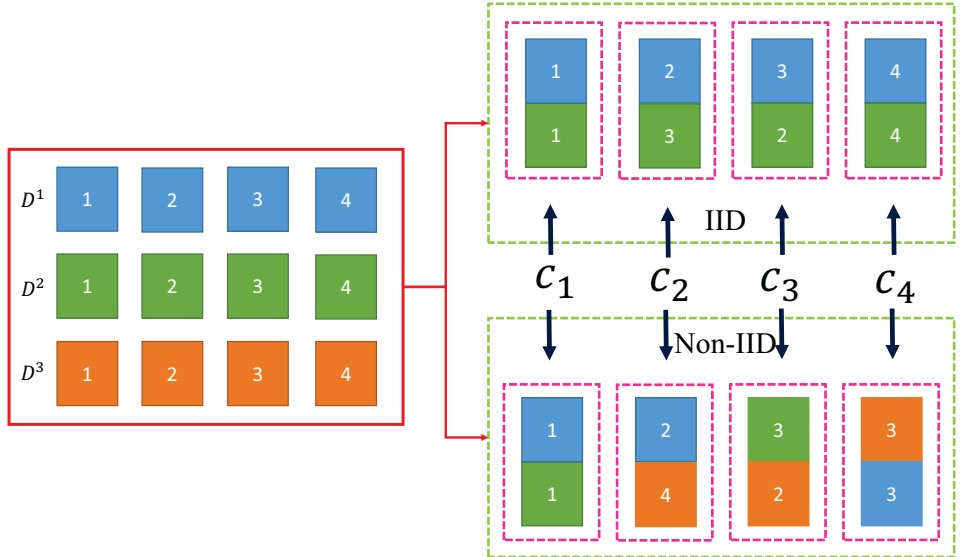

Figure 3: Non-IID and IID datasets generation example

2. For IID dataset, determine the number of classes each client as $k_c$. Sample $k_c$ classes from the $K$ classes, and denote the sampled class sets as $K_{sample}$, $|K_{sample}| = k_c$. For each $i$, $i \in K_{sample}$, assign each data partition $D_n^i$, $n = 1, \ldots, N$, to each client without overlapping. Goto step 4.

3. For non-IID dataset, $P_i(X)$, the distribution of feature $X$, on each client $i$ may not be the same. Here the difference from step 2 is that in non-IID sampling, each client will independently sample a number of $k_c$ classes. The rest operations are the same as step 2.

4. Dataset on each client is partitioned randomly into training and testing sets. And two sets are IID for each client.

When the number of clients is $N = 4$, the total number of classes is $K = 3$, the total number of classes for each client is $k_c = 2$, the generation of IID and non-IID is shown in Fig. 3 (Note for non-IID dataset generataion, $C_2$ and $C_4$ are IID by chance).

## C  ANALYSIS OF PERSONALIZED ALGORITHM

We chose the CNN model architecture in the following table 4. In the experiment, the accuracy rate

Table 4: Convolutional network model structure

| Conv $(5 \times 5 \times 20)$ |
| :---: |
| MaxPool $(2 \times 2)$ |
| Conv $(5 \times 5 \times 50)$ |
| MaxPool $(2 \times 2)$ |
| Full Connect (512) |
| Full Connect (Number of categories in the data set) |

is used as the evaluation index on each client, as shown in the formula (15), where TP, TN, FP, and FN represent true positive, true negative, false positive, and false negative.

$$acc = \frac{TP + TN}{TP + TN + FP + FN}. \tag{15}$$

After confirming the experimental parameters settings, we will conduct comparative experiments with the parameters settings in Table 5. The method used in the experiment is the federated average learning algorithm (FedAvg). Table 6 is the corresponding experimental result.

Table 5: Federated average algorithm exploration experiment parameters table

| Number | datasets | Data distribution | Training way | Total client | train client | categories | train | test |
| :---: | :---: | :---: | :---: | :---: | :---: | :---: | :---: | :---: |
| 3-1-1 | MNIST | IID | Independent | 10 | 10 | 6 | 2800 | 700 |
| 3-1-2 | MNIST | IID | Federated | 10 | 10 | 6 | 2800 | 700 |
| 3-1-3 | MNIST | non-IID | Independent | 10 | 10 | 6 | 2800 | 700 |
| 3-1-4 | MNIST | non-IID | Federated | 10 | 10 | 6 | 2800 | 700 |
| 3-1-5 | EMNIST | IID | Independent | 10 | 10 | 32 | 6100 | 1500 |
| 3-1-6 | EMNIST | IID | Federated | 10 | 10 | 32 | 6100 | 1500 |
| 3-1-7 | EMNIST | non-IID | Independent | 10 | 10 | 32 | 6100 | 1500 |
| 3-1-8 | EMNIST | non-IID | Federated | 10 | 10 | 32 | 6100 | 1500 |
| 3-1-9 | CIFAR10 | IID | Independent | 10 | 10 | 6 | 2400 | 600 |
| 3-1-10 | CIFAR10 | IID | Federated | 10 | 10 | 6 | 2400 | 600 |
| 3-1-11 | CIFAR10 | non-IID | Independent | 10 | 10 | 6 | 2400 | 600 |
| 3-1-12 | CIFAR10 | non-IID | Federated | 10 | 10 | 6 | 2400 | 600 |
| 3-1-13 | CIFAR100 | IID | Independent | 10 | 10 | 60 | 2400 | 600 |
| 3-1-14 | CIFAR100 | IID | Federated | 10 | 10 | 60 | 2400 | 600 |
| 3-1-15 | CIFAR100 | non-IID | Independent | 10 | 10 | 60 | 2400 | 600 |
| 3-1-16 | CIFAR100 | non-IID | Federated | 10 | 10 | 60 | 2400 | 600 |

The experimental results show that the results of the FL algorithm are often better than the independent training method when the data is IID. The result of the FL will be worse than that of the independent training when the data is non-IID. This experimental conclusion shows that the federated average learning algorithm (FedAvg), which executes the average aggregation step of the model parameters , is more suitable for the case where the datasets on the client are IID. It has the opposite effect when the data is non-IID. To further explore the influence of the federated averaging algorithm on the model parameters under non-IID, we calculated the average value of each client's upload gradient similarity in different distribution scenarios. The results are shown in Fig. 4 and Fig. 5. Based on the above experimental results, we can find that: (1) In independent and synchronized steps, the mean of the gradient similarity is close to 1. In a non-IID scenario, the mean of the

Table 6: Federal average experimental results (percentage%)

| Number | c1 | c2 | c3 | c4 | c5 | c6 | c7 | c8 | c9 | c10 | Average accuracy |
|--------|------|------|------|------|------|------|------|------|------|------|------------------|
| 3-1-1 | 95.57 | 97.92 | 97.23 | 97.64 | 97.09 | 97.23 | 97.23 | 97.51 | 97.51 | 98.20 | 97.313 |
| 3-1-2 | **97.09** | **98.47** | **98.47** | **98.75** | **98.78** | **98.34** | **98.75** | **98.61** | **98.34** | **98.47** | **98.307** |
| 3-1-3 | 94.60 | 97.45 | **98.23** | **97.10** | 96.99 | 97.49 | **98.03** | 97.30 | 96.95 | 97.72 | **97.186** |
| 3-1-4 | **95.57** | **97.87** | 97.55 | 96.41 | **97.40** | **97.49** | 96.76 | 96.74 | 96.95 | **98.57** | 97.131 |
| 3-1-5 | 87.04 | 85.61 | 84.76 | 84.30 | 84.40 | 85.54 | 84.96 | 84.50 | 84.71 | 85.67 | 85.095 |
| 3-1-6 | **90.59** | **90.95** | **90.36** | **90.23** | **89.12** | **90.69** | **90.55** | **89.64** | **89.58** | **91.47** | **90.318** |
| 3-1-7 | **86.71** | 82.22 | 87.17 | **84.11** | 84.76 | **82.68** | **84.17** | 82.16 | 84.89 | **82.16** | 84.103 |
| 3-1-8 | 85.80 | **82.87** | **89.51** | 82.35 | **85.09** | 81.96 | 83.85 | **83.65** | **87.63** | 79.68 | **84.239** |
| 3-1-9 | 60.50 | 64.50 | 69.50 | 63.66 | 63.33 | 63.50 | 62.83 | 65.16 | 63.00 | 59.66 | 63.564 |
| 3-1-10 | **67.00** | **69.83** | **74.66** | **72.66** | **69.50** | **71.33** | **69.83** | **71.00** | **69.50** | **68.16** | **70.347** |
| 3-1-11 | **60.50** | **58.33** | 56.16 | **49.83** | **62.16** | **62.50** | **63.33** | **59.66** | 57.00 | **64.83** | **59.43** |
| 3-1-12 | 56.83 | 46.83 | **59.16** | 41.83 | 61.16 | 62.16 | 61.50 | 46.33 | **57.50** | 57.00 | 55.03 |
| 3-1-13 | 19.66 | 22.5 | 20.5 | 20.66 | 21.16 | 20.83 | 21.5 | 18.83 | 23.83 | 21 | 21.047 |
| 3-1-14 | **32.33** | **34.16** | **33.5** | **30.66** | **32.83** | **32.66** | **30.33** | **27.5** | **33.16** | **28** | **31.513** |
| 3-1-15 | 20 | 18.5 | 17.83 | 16.66 | 20 | 19.16 | 18.16 | 17 | 18.16 | 17 | 18.247 |
| 3-1-16 | **22.66** | **20.66** | **20.16** | **18.16** | **20.33** | **19.5** | **19.83** | **20** | **21** | **19** | **20.13** |

gradient similarity on the client differs. (2) In the non-IID, compared with the deeper layer features, the mean of the SNGS will be higher for the shallow layer features.

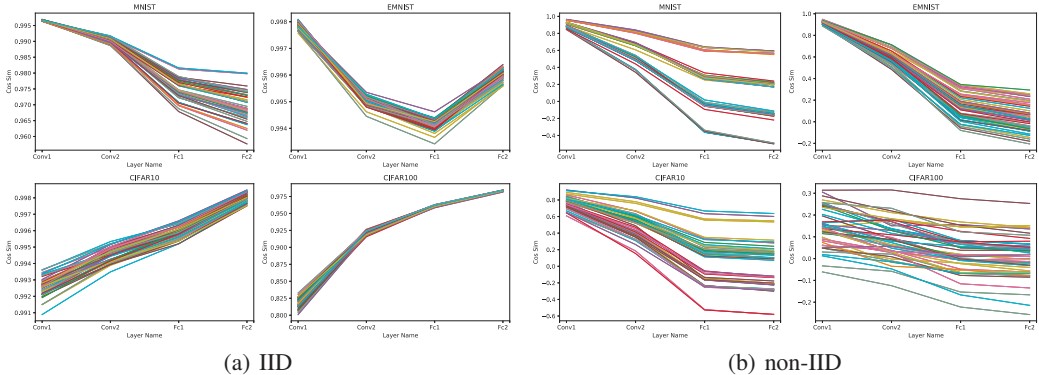

(a) IID  (b) non-IID

Figure 4: Gradient similarity statistics under MNIST, EMNIST, CIFAR10, CIFAR100 for each client category number of 4, 32, 4, 40

## D ANALYSIS OF HYPERPARAMETERS ON PERSONALIZED EXPERIMENTS

We explored the impact of update frequency of the hyperparameters similarity matrix $\Gamma$, the number of classes $k$, the number of clients $N$, and the different local learning rounds $E$ on the experimental results, which are shown in Table. 7.

## E FINE-TUNE EXPERIMENTAL RESULTS

When the personalized FL algorithm is completed, the server delivers the final personalized model to each client. Like the meta-learning algorithm, after each client receives a proprietary personalized model, it can still make fine-tuning on itself datasets based on the model to achieve better results. The primary considerations for fine-tuning are the learning rate ($\beta$), the training batch size ($B$), and the number of steps the fine-tuning takes (step).

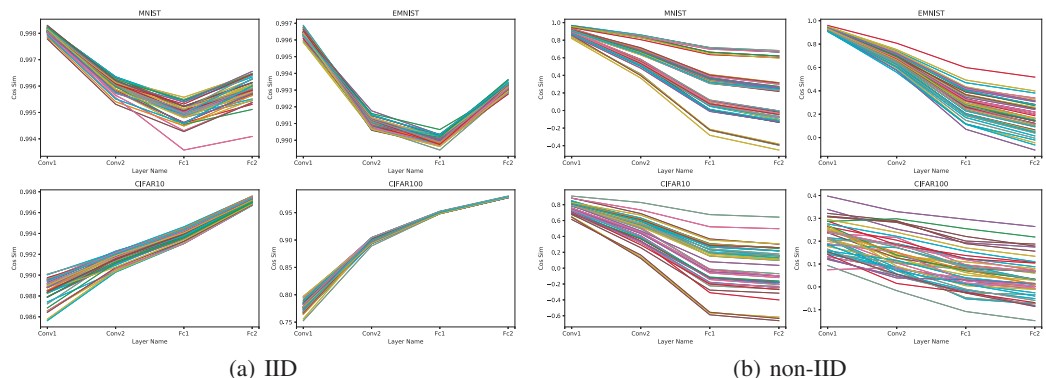

(a) IID                                                    (b) non-IID

Figure 5: Gradient similarity statistics under MNIST, EMNIST, CIFAR10, CIFAR100 for each client of class number of 6, 40, 6, 60

Table 7: The influence of hyperparameters on the experiment of personalized federated algorithm

| Methods | MNIST | | CIFAR10 | | EMNIST | | CIFAR100 | |
|---|---|---|---|---|---|---|---|---|
| | IID | non-IID | IID | non-IID | IID | non-IID | IID | non-IID |
| T = 5 | 98.86 | 98.32 | 76.06 | 68.18 | 90.67 | 88.58 | 38.20 | 31.36 |
| T = 10 | 98.82 | 98.44 | 75.76 | 68.91 | 90.58 | 88.76 | 37.06 | 30.94 |
| T = 20 | 98.65 | 98.30 | 75.15 | 68.48 | 90.13 | 88.62 | 35.4 | 29.38 |
| k1 | 98.96 | 98.32 | 82.05 | 73.3 | 91.23 | 90.14 | 42.2 | 31.2 |
| k2 | 98.82 | 98.44 | 75.76 | 68.91 | 90.58 | 88.76 | 37.06 | 30.94 |
| k3 | 98.75 | 98.30 | 68.03 | 66.55 | 87.88 | 87.21 | 31.65 | 30.41 |
| N = 6 | 98.70 | 98.01 | 75.25 | 66.88 | 89.74 | 88.65 | 33.41 | 28.08 |
| N = 8 | 98.87 | 98.24 | 75.64 | 67.99 | 90.33 | 88.53 | 35.99 | 30.10 |
| N = 10 | 98.82 | 98.44 | 75.76 | 68.91 | 90.58 | 88.76 | 37.06 | 30.94 |
| E = 1 | 98.82 | 98.44 | 75.76 | 68.91 | 90.58 | 88.76 | 37.06 | 30.94 |
| E = 2 | 99.05 | 98.79 | 77.16 | 72.23 | 90.71 | 88.98 | 37.84 | 30.63 |
| E = 5 | 99.18 | 98.76 | 77.11 | 69.83 | 90.66 | 88.56 | 36.68 | 28.99 |

Table. 8 shows the model based on SPFL training, according to consideration factors, the results of multiple sets of different experimental parameters . First, we compared the accuracy of the model after fine-tuning with the previous model. It can be found that fine-tuning can indeed improve the performance of the model. However, compared with the improvement of SPFL in the federated average algorithm, the benefits of fine-tuning are not very good. It may because that the model after learning has original personalized attributes.

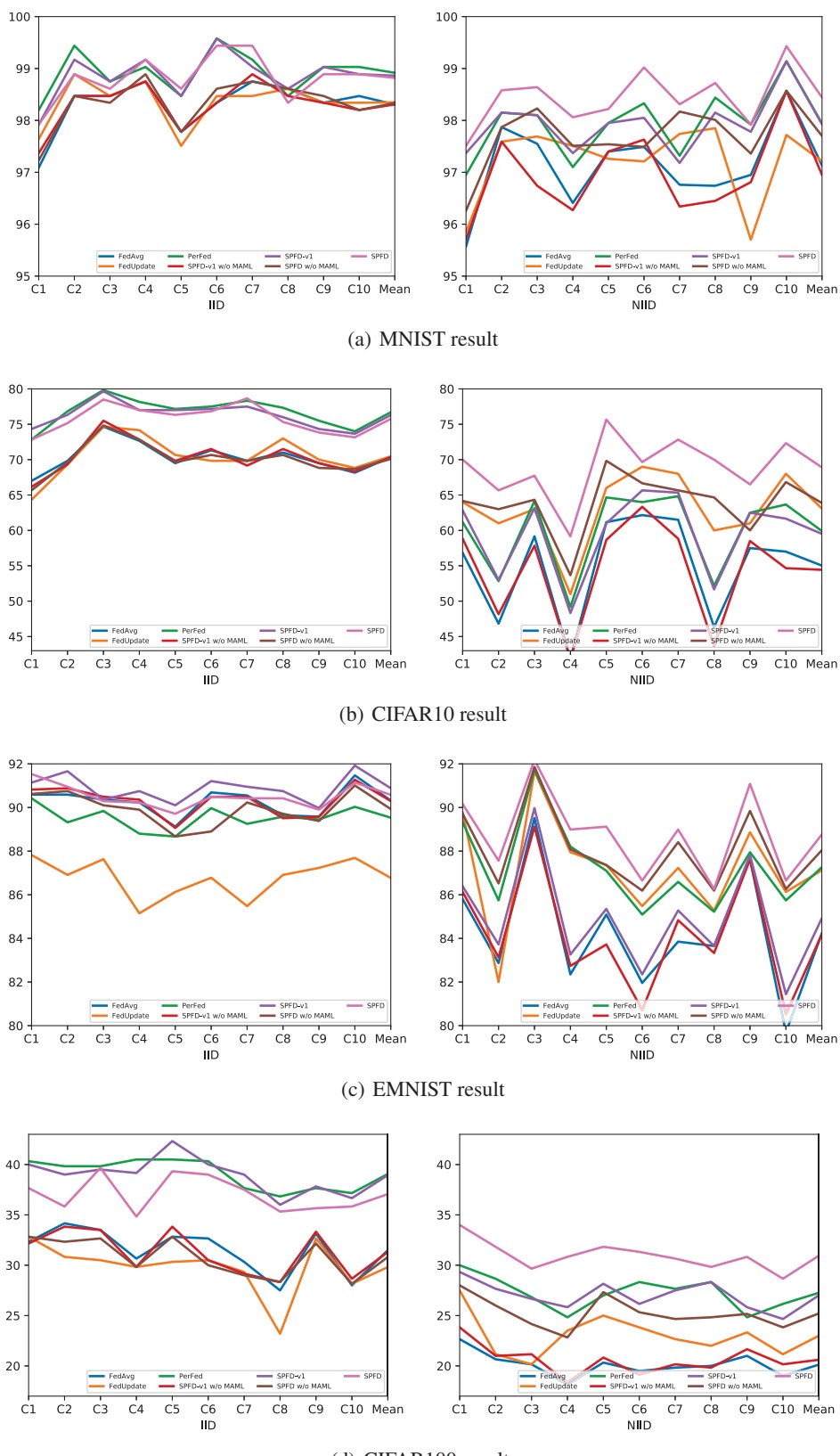

(a) MNIST result

(b) CIFAR10 result

(c) EMNIST result

(d) CIFAR100 result

Figure 6: The results of each client of the SPFL algorithm on the four datasets

Table 8: SPFL algorithm fine-tuning experimental results

| datasets | distributed | B=128 $\beta$=e-2 step=10 | B=128 $\beta$=e-3 step=10 | B=128 $\beta$=e-4 step=10 | B=128 $\beta$=e-3 step=20 | B=128 $\beta$=e-3 step=100 | B=64 $\beta$=e-3 step=10 |
|---|---|---|---|---|---|---|---|
| MNIST | IID | 98.57 | 98.85 | 98.82 | 98.86 | **98.89** | 98.87 |
| | non-IID | 98.20 | 98.44 | 98.45 | 98.48 | 98.47 | **98.49** |
| CIFAR10 | IID | 73.75 | 75.91 | 75.8 | **76.08** | 76.01 | 75.98 |
| | non-IID | 65.66 | 69.14 | 69.21 | 69.14 | **69.41** | 69.31 |
| EMNIST | IID | 89.62 | 90.40 | 90.53 | 90.42 | **90.43** | 90.35 |
| | non-IID | 87.56 | 88.82 | 88.80 | 88.76 | **88.85** | 88.77 |
| CIFAR100 | IID | 32.93 | **37.55** | 37.29 | 37.30 | 37.30 | 37.33 |
| | non-IID | 26.16 | **31.68** | 31.13 | 31.18 | 31.18 | 31.21 |