# OpenReview forum: "Personalized Heterogeneous Federated Learning with Gradient Similarity"
_ICLR.cc/2022/Conference — ICLR 2022 Submitted_

### Official Review · Reviewer_3fbQ · 2021-11-02

**Correctness:** 3
**Technical Novelty And Significance:** 2
**Empirical Novelty And Significance:** 2
**Recommendation:** 3
**Confidence:** 4

**Main Review:**

1. The proposed method requires modeling the pair-wise similarity between each client. This requires maintaining a potential large matrix, maintains the identity of each client, and cannot be applied to new clients.
2. The algorithms are all tested on synthetic (label-heterogeneous) Federated dataset. Could you test on natural heterogeneous dataset?

The paper is overall hard to read due to typos, grammar errors, and influent sentences. Here are some examples I found on page 4 and 5:

- "Then the SGD updates of the corresponding gradients of $w_i^t$ and $w_j^t$ are:"
- "where ∥ · ∥ is the vector L 2normal. And the aggregation strategy on the server for C iis as follows:"
- This section introduces an asynchronous updating strategy that addresses the system heterogeneity problem mentioned in Section [missing].

These aspects make the current version of this paper look incomplete.

**Summary Of The Paper:**

This paper proposed two personalized federated learning algorithms: SPFL, PLGA.

**Summary Of The Review:**

The paper is overall not well-written and I recommend rejection.

---

### Official Review · Reviewer_7GZS · 2021-11-03

**Correctness:** 2
**Technical Novelty And Significance:** 2
**Empirical Novelty And Significance:** 2
**Recommendation:** 3
**Confidence:** 4

**Main Review:**

There is no question that personalized federated learning is an important and practical problem. The proposed approaches and initial results presented seem promising, but they are very far from solving the problem.

For a paper proposing new methods for Federated Learning, one generally expects either that theoretical convergence analysis is provided, or a very thorough experimental evaluation is provided. This paper doesn't provide any theoretical guarantees (and the motivation for the proposed approaches is not always clear). There are a lot of questions about the experiments, which reduce my confidence in the overall conclusions. Moreover, the proposed schemes appear to be fundamentally incompatible with privacy, which is a critical factor for Federated Learning methods. Consequently, I don't believe that this paper is acceptable. Substantial additional work would need to be done (experiments and/or theory) to improve it to an acceptable state. Detailed comments follow.

The paper is missing references to other relevant and more recent work on asynchronous FL:
* Chen et al., "Asynchronous Online Federated Learning for Edge Devices with Non-IID Data,"  arXiv:1911.02134
* van Dijk et al., "Asynchronous Federated Learning with Reduced Number of Rounds and with Differential Privacy from Less Aggregated Gaussian Noise," arXiv:2007.09208
* Chai et al., "A Communication-Efficient Federated Learning Method with Asynchronous Tiers Under Non-IID Data," arXiv:2010.05958
* Nguyen et al., "Federated Learning with Buffered Asynchronous Aggregation," arXiv:2106.06639

I'm confused by the description between equations (3) and (4). Does the server update the client parameters $w_i^{t+1}$ and $w_j^{t+1}$?

The proposed scheme either requires clients to directly share model updates $g_i^t$ with each other, or they need to share their individual model updates directly with the server. This violates the general principle in Federated Learning that the server should only receive/process updates aggregated from many clients, in order to better preserve the privacy of client models. Hence, the proposed scheme seems fundamentally incompatible with current practices in Federated Learning systems. Is there a way to preserve privacy while also performing these types of updates?

Additional explanation should be included to justify how (10) alleviates the update gap between stragglers and the server. In addition, it would be useful to illustrate the impact of using (10) vs. another approach experimentally. In order to implement the similarity calculation (8) at the server, the server must maintain a long history of the past model versions (up to some maximum staleness). Is this additional memory overhead practical? And again, it does not appear to be possible to perform such similarity-weighted updates in a privacy-preserving manner.

The methods, such as PLGA, are described without very much motivation and no formal justification (e.g., no theoretical convergence guarantees, even under strong assumptions).

The experiments appear to fix some of the hyperparameters to be the same for all of the methods compared (e.g., local and server learning rates). Normally each method will benefit from having these tuned separately per-method. Because of this, I'm not convinced that the comparison is a fair one.

How many clients are there in the SPFL evaluation?

**Summary Of The Paper:**

This paper proposes two methods for personalized federated learning, one synchronous and one asynchronous. The general approach taken in both cases is to adapt the weights when averaging information from different clients, so that clients with more similar gradients are given more weight in the update for each client. The two approaches are called SPFL (synchronous) and PLGA (asynchronous). The approaches are illustrated on small image classification tasks.

**Summary Of The Review:**

For a paper proposing new methods for Federated Learning, one generally expects either that theoretical convergence analysis is provided, or a very thorough experimental evaluation is provided. This paper doesn't provide any theoretical guarantees (and the motivation for the proposed approaches is not always clear). There are a lot of questions about the experiments, which reduce my confidence in the overall conclusions. Moreover, the proposed schemes appear to be fundamentally incompatible with privacy, which is a critical factor for Federated Learning methods. Consequently, I don't believe that this paper is acceptable.

---

### Official Review · Reviewer_1psQ · 2021-11-08

**Correctness:** 3
**Technical Novelty And Significance:** 2
**Empirical Novelty And Significance:** 3
**Recommendation:** 3
**Confidence:** 4

**Main Review:**

Strength: The idea behind the paper is intuitive, and the numerical results also show its advantage compared to other methods.

Weaknesses:

- My first concern is regarding the computational cost of the method. Note that classic FedAvg, let's say with full participation, requires $\mathcal{O}(Nd)$ computation at each iteration, where $N$ is the number of clients and $d$ is dimension. However, for this method, the computation would be $\mathcal{O}(N^2 d)$, since all cross similarities are being computed.

- My second concern is on the memory budget that this method requires on the server-side (especially for the synchronous part). Does the server need to store all local gradients to compute the update? And how feasible is this in practice?

- My third concern is on the sensitivity of this method to gradient accuracy. In many cases, especially in real-world applications, the local gradients might be very noisy. For instance, they are computed over small batches or because noise is added intentionally to satisfy privacy constraints. It seems to me that this method might not work properly in such settings. I wonder whether the authors have considered this matter or not.


**Summary Of The Paper:**

This paper proposed a personalized federated learning algorithm which takes into account the similarity of gradient of different users to update the model. More formally, the authors define $\tilde{S}(i,j)$ as a measure of similarity between the gradients of two user $i$ and $j$, and then update the model of user $i$by weighting the gradient of user $j$ by $\tilde{S}(i,j)$. The authors study their method in various numerical settings.

**Summary Of The Review:**

I find the idea of the paper intuitive and interesting. However, I have a number of concerns about the computation and memory cost of the method and the sensitivity of the proposed approach to the exactness of gradients.
I am open to raising my score, given that my concerns are (at least partially) addressed.

---

### Official Review · Reviewer_HSpu · 2021-11-08

**Correctness:** 3
**Technical Novelty And Significance:** 3
**Empirical Novelty And Significance:** 4
**Recommendation:** 5
**Confidence:** 3

**Main Review:**

*Strengths*
1. Interesting idea of weighted averaging of client gradients to do personalized FL
2. Potentially useful experimental results.

*Weaknesses*
1. Table 1: Factoring in standard error, it is not clear if there is a clear gain for their algorithm except in CIFAR-10 with non-IID setting. So the bolding does not make sense. However, I am not saying that only positive results should be published. Negative results are also good, as long as the experiments and methods are well thought out.

2. Since this is a experimental paper, author could augment their contributions with more comprehensive and useful experiments.

3. Can the authors please add error bars to the table?

4. What is “stage strategy?” This is never defined nor discussed.

5. Table 2: What is client and server?

6. Typo: straggers

7. Typo: Where is “Diag” used? There seems to be some mistake in eq (10). I don’t understand why (10) or (11) has this form. Please put more effort in explaining your strategy and the intuition behind it. Currently it is hard to follow.


*Comments*
1. Eqn (3): Could enhance the similarity metric S(i,j) by a weighting hyperparamter \gamma. S(i,j) = \gamma * <g_i, g_j>/(\|g_i\| \|g_j\|).

2. Are the authors reporting standard errors or standard deviations in the table?

3. Sec 4.1: What was the reasoning behind fixing the common parameters at these particular values?

4. It is not fair that authors provide a non-anonymous github link.

**Summary Of The Paper:**

This paper proposes gradient based weighting strategies for synchronous (SPFL) and asynch (PLGA) personalized Federated Learning (FL). Authors also provide experimental results comparing their methods to baselines on many benchmark datasets.


**Summary Of The Review:**

Although the main idea seems promising, there needs to be a lot work done on presentation and evaluation.

---

> ### Comment · Reviewer_HSpu · 2021-11-26
> **Evaluation remains the same after discussion period**
>
> Evaluation remains the same as there were no further revisions or discussion.

---

### Decision · Program_Chairs · 2022-01-20

**Decision:**

Reject

**Comment:**

This paper proposed a personalized federated learning algorithm which takes into account the similarity of gradient of different users to update the model. Although the ideas presented are intuitive, the algorithms have fundamental limitations, for example, they may cause large overhead of memory, communication and computation, and are unsuitable for privacy-preserving machine learning. In addition, there are no rigorous analysis and the experiments are not convincing. This is a clear rejection.